# Fractional Dynamical Systems Solved by a Collocation Method Based on Refinable Spaces

Laura Pezza [1,*,†] and Simmaco Di Lillo [2,*,†]

1   Department of Basic and Applied Sciences for Engineering (SBAI), Università di Roma "La Sapienza",
    Via A. Scarpa 16, 00161 Rome, Italy
2   Department of Mathematics Guido Castelnuovo, Università di Roma "La Sapienza", Piazzale Aldo Moro 5,
    00185 Rome, Italy
*   Correspondence: laura.pezza@sbai.uniroma1.it (L.P.); dilillo.2016774@studenti.uniroma1.it (S.D.L.)
†   These authors contributed equally to this work.

**Abstract:** A dynamical system is a particle or set of particles whose state changes over time. The dynamics of the system is described by a set of differential equations. If the derivatives involved are of non-integer order, we obtain a fractional dynamical system. In this paper, we considered a fractional dynamical system with the Caputo fractional derivative. We collocated the fractional differential problem in dyadic nodes and used refinable functions as approximation functions to achieve a good degree of freedom in the choice of the regularity. The collocation method stands out as a particularly useful and attractive tool for solving fractional differential problems of various forms. A numerical result is presented to show that the numerical solution fits the analytical one very well. We collocated the fractional differential problem in dyadic nodes using refinable functions as approximation functions to achieve a good degree of freedom in the choice of regularity.

**Keywords:** fractional differential problem; collocation method; fractional derivative; B-spline

**MSC:** 00A69

## 1. Introduction

For a long time, fractional calculus was considered to be a purely mathematical field with no real applications. Recently, however, this has changed. Think, for example, of all the problems where it is important to describe the behaviour and evolution of the function over a period of time and not just at a specific point in time. Think, for example, of materials with memory. Fractional calculus is, therefore, particularly important for studying and solving this type of problem.

Many researchers have found that fractional derivatives are suitable for mathematical modelling of physical problems such as diffusion in biological tissues [1–3], propagation in porous media [4], anomalous diffusion [5–7], viscoelasticity [8–11], and earthquakes [12]. Fractional calculus has also been considered for pandemic models [13–15] or in finance [16].

Since the fractional derivate involves an integral, many authors use quadrature formulae or discrete derivatives [17] to approximate the fractional derivative. Others use Galerkin methods [18,19] or spectral methods [20].

In our previous works [21,22], we proved that if we collocate the fractional differential problem in dyadic nodes and we use the B-splines as approximation functions, we obtain a collocation method that works very well. Moreover, the method takes advantage of the refinability and derivability properties of the B-splines in order to produce an accurate and efficient algorithm. Indeed, the fractional derivatives of the B-splines appearing in the collocation matrix satisfy a closed form involving fractional B-splines of lower orders. In this paper, we used this collocation method, generalised to a wider class of approximating refinable functions [23], and we used this method to numerically solve

a fractional dynamical system. By using these fractional functions instead of fractional B-splines, we have more degrees of freedom in the choice of the regularity. For all these reasons, the collocation method stands out as a particularly useful and attractive tool for the solution of fractional differential problems of different forms [24].

The paper is structured as follows: In the first section, we describe the fractional dynamical systems and we define the fractional derivatives in several meaningful ways. In the second section, fractional refinable functions are introduced, as well as their main properties. The most-important property in the context of this paper, i.e., the fractional differential rule, is treated in the third section, as well as the way to differentiate the left-edge B-splines in the fractional sense. In the following section, a Multiresolution Analysis (MRA) on the semi-bounded interval $[0, +\infty)$ is introduced. The fractional collocation method is then introduced, specifying the entries of the collocation matrix, and a numerical test is provided in the last section. We compare one of the other results obtained using several B-splines and fractional B-splines in [18,21,25,26].

## 2. Fractional Dynamical Systems

Consider the following fractional dynamical system:

$$\begin{cases} D_t^\gamma X(t) = A(t)\, X(t), & \gamma \in (0,1),\ t > 0 \\ X(0) = X_0 \end{cases} \tag{1}$$

where $X(t) = (x_1(t), x_2(t), \ldots, x_n(t)) \in \mathbb{R}^n$, and $A(t) \in \mathbb{R}^{n \times n}$.

In this context, $D_t^\gamma y$ denotes the Caputo fractional derivative with respect to the time $t$ defined as

$$D_t^\gamma\, y(t) := \left( \mathcal{J}^{(k-\gamma)} y^{(k)} \right)(t), \quad \gamma > 0 \tag{2}$$

where $k$ is an integer such that $k - 1 < \gamma < k$ and $\mathcal{J}^{(\gamma)}$ is the Riemann–Liouville integral operator given by

$$\left( \mathcal{J}^{(\gamma)} y \right)(t) := \frac{1}{\Gamma(\gamma)} \int_0^t y(\tau)\, (t - \tau)^{\gamma - 1}\, d\tau, \quad t > 0.$$

In the above equation, $\Gamma$ denotes Euler's gamma function:

$$\begin{cases} \Gamma(\gamma) = \int_0^{+\infty} x^{\gamma - 1} e^{-x}\, dx & \text{for } \gamma > 0, \\ \Gamma(\gamma) = \gamma^{-1} \Gamma(\gamma + 1) & \text{for } \gamma < 0. \end{cases}$$

Through the Gamma function, it is possible to define the generalised binomial coefficients:

$$\binom{\gamma}{l} = \frac{\Gamma(\gamma + 1)}{\Gamma(l + 1)\Gamma(\gamma - l + 1)}, \quad l \in \mathbb{Z}. \tag{3}$$

If $\gamma$ is a positive integer, the above definition and the usual definition of the binomial coefficient coincide, while they are infinitely supported if $\gamma \in \mathbb{R}^+ \backslash \mathbb{N}$ or zero if $l \in \mathbb{Z} \backslash \mathbb{N}$.

However, they decay to infinity as [27]

$$\binom{\gamma}{l} = O(l^{-\gamma - 1}) \quad \text{for} \quad l \to \infty.$$

We note that, when the homogeneous initial conditions are imposed on the function $y(t)$, the Caputo definition (2) coincides with the Riemann–Liouville definition:

$$D_t^\gamma\, y(t) := \frac{d^k}{dt^k} \left( \mathcal{J}^{(\gamma)} y \right)(t), \quad t > 0,$$

which requires less regularity on $y(t)$, and both reduce to the usual differential operator in the case of $\gamma \in \mathbb{N}$ [28].

In the following, we will make use of $D_t^{\gamma} f(2^j t)$, and so, we will need the following theorem, already proven in [22].

**Theorem 1.** *Let $\gamma$ be a real number such that $0 < \gamma < 1$ then*

$$D_t^{\gamma} f(2^j t) = 2^{j\gamma} D_{(2^j t)}^{\gamma} f(2^j t).$$

**Proof.** Denote $F(t) = f(2^j t)$, then $F^{(m)}(t) = 2^{jm} f^{(m)}(2^j t)$, $m \in \mathbb{N}$. Therefore,

$$D_t^{\gamma} F(t) = \frac{1}{\Gamma(k-\gamma)} \int_0^t \frac{F^{(k)}(\tau)}{(t-\tau)^{(\gamma-k+1)}} d\tau =$$

$$= \frac{1}{\Gamma(k-\gamma)} 2^{jk} \int_0^t \frac{f^{(k)}(2^j \tau)}{(t-\tau)^{(\gamma-k+1)}} d\tau.$$

After changing the variable in the integral, we obtain

$$D_t^{\gamma} F(t) = \frac{1}{\Gamma(k-\gamma)} 2^{jk} \int_0^{2^j t} \frac{f^{(k)}(\tau)}{(t-2^{-j}\tau)^{(\gamma-k+1)}} 2^{-j} d\tau =$$

$$= \frac{1}{\Gamma(k-\gamma)} \frac{2^{jk} 2^{-j}}{2^{-j(\gamma-k+1)}} \int_0^{2^j t} \frac{f^{(k)}(\tau)}{(2^j t-\tau)^{(\gamma-k+1)}} d\tau = 2^{j\gamma} D_{(2^j t)}^{\gamma} f(2^j t).$$

$\square$

In the next section, we introduce a new collocation method, the multi-scale collocation method, which takes advantage of multi-scale techniques to produce an efficient and accurate algorithm.

## 3. Fractional Cardinal B-Splines and Fractional GP Functions

In the paper [27], the extension of the class of the cardinal B-splines to the fractional ones is provided by introducing the concept of the fractional finite difference operator. The starting point is the definition of the classical cardinal B-spline, i.e., the B-spline on integer knots. Consider the truncated power function:

$$T^n(x) := (\max(0, x))^n, \quad x \in \mathbb{R}, \quad n \in \mathbb{N}$$

and the finite difference operator:

$$\Delta^n f(x) := \sum_{k=0}^n \binom{n}{k} (-1)^k f(x-k), \quad x \in \mathbb{R}.$$

Then, the classical B-spline of integer-order $n$ is defined by the formula:

$$B^n(x) := \frac{1}{(n+1)!} \Delta^{n+1} T^n(x). \tag{4}$$

From this definition, the Fourier transform of $B^n$ becomes

$$\hat{B}^n(\omega) = \left( \frac{1 - e^{-i\omega}}{i\omega} \right)^{n+1}.$$

Due to the definition of generalised binomial coefficients, the definition (4) can be extended to a real index $\alpha$. Hence, using the fractional finite difference operator:

$$\Delta^{\gamma} f(x) := \sum_{k \geq 0} \binom{\gamma}{k} (-1)^k f(x-k),$$

we can define the fractional B-spline of order $\alpha$:

$$B^\alpha(x) := \frac{1}{\Gamma(\alpha+1)} \Delta^{\alpha+1} T^\alpha(x).$$

It is also easy to prove that

$$\Delta^\gamma B^n(x) = \frac{\Delta^{n+1} T^{n-\gamma}(x)}{\Gamma(n-\gamma+1)}.$$

Thus, due to the finite fractional differential rule [22]:

$$D^\gamma B^n(x) = \Delta^\gamma B^{n-\gamma}(x) \tag{5}$$

we can say that the fractional derivative of a B-spline is a fractional B-spline.

It easy to prove that the Fourier transform of $B^\alpha$ becomes

$$\hat{B}^\alpha(\omega) = \left(\frac{1-e^{-i\omega}}{i\omega}\right)^{\alpha+1}.$$

A similar theory can be developed for the fractional derivatives of a class of refinable functions compactly supported in $[0, n+1]$, called Gori–Pitolli (GP) refinable functions, depending on a real parameter $h \geq n$. These functions are denoted by $\varphi_{n,h}$. Their fractional derivatives are fractional GP refinable functions, i.e., GP refinable functions with non-integer-order $\alpha$: $\varphi_{\alpha,h}$ (see Figure 1). See [23,25,26,29] for more details.

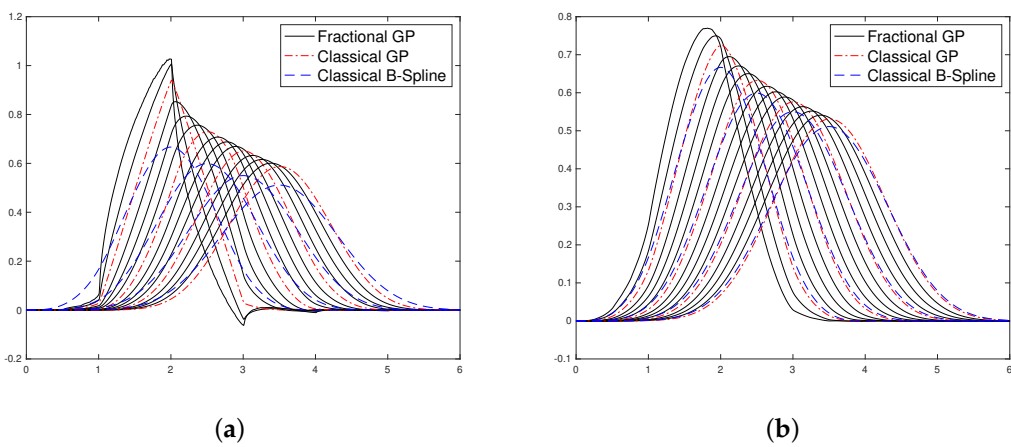

**(a)**             **(b)**

**Figure 1.** (**a**) The graphs of the fractional GP for $\alpha = 2{:}0.25{:}6$, $h = \alpha + 3$, classical GP for $\alpha = 2, \ldots, 6$, and classical B-splines for $h = \alpha$ integer. (**b**) The graphs of the fractional GP for $\alpha = 2{:}0.25{:}6$, $h = \alpha + 0.5$, classical GP for $\alpha = 2, \ldots, 6$, and classical B-splines for $h = \alpha$ integer.

In Figure 1a, the fractional GP functions are plotted for $\alpha = 2{:}0.25{:}6$ and $h = \alpha + 3$. It is possible to see the continuous dependence of the family on the index $\alpha$ and the interpolation with the fractional B-splines of [27] when $\alpha = h$.

Instead, when $h >> \alpha$, each B-spline looks like the corresponding one of degrees less than 2.

The same observations can be made for Figure 1b, where $h = \alpha + 0.5$; here, the fractional functions are more similar to the fractional B-splines.

In both cases, the support of the fractional GP function looks like $[0, \sigma]$, where $\sigma = \lceil \gamma \rceil + 1$.

### 4. Multiresolution Analysis on $\mathbb{R}$ and on $[0, \infty)$

Let $\{V_j, \ j \in \mathbb{Z}\}$ be a sequence of closed subspaces of $L^2(\mathbb{R})$. The sequence forms a multiresolution analysis (MRA) of $L^2(\mathbb{R})$ if:

(i)　$V_j \subset V_{j+1}, \ j \in \mathbb{Z}$;

(ii)　$\displaystyle\bigcup_{j \in \mathbb{Z}} V_j = L^2(\mathbb{R})$;

(iii)　$\displaystyle\bigcap_{j \in \mathbb{Z}} V_j = \{0\}$;

(iv)　$f(t) \in V_j \leftrightarrow f(2t) \in V_{j+1}, j \in \mathbb{Z}$;

(v)　There exists an $L^2(\mathbb{R})$-stable basis in $V_0$.

An MRA can be generated by a refinable function, i.e., a function defined through a refinement mask $a = \{a_k \in \mathbb{R}, \ k \in \mathbb{Z}\}$ and a refinement equation, as follows:

$$\varphi(t) = \sum_{k \in \mathbb{Z}} a_k \, \varphi(2\,t - k), \qquad t \in \mathbb{R}. \tag{6}$$

Suitable conditions on the mask coefficients $\{a_k\}$ ensure the existence of a unique function $\varphi$ solution to (6), which belongs to $L^2(\mathbb{R})$ and such that their integer translates $\{\varphi(t - k), k \in \mathbb{Z}\}$ form an $L^2(\mathbb{R})$-stable basis in $V_0$ (see [30] for details). As a consequence, the refinable function $\varphi$ generates all the spaces $V_j$ through dilation and translation, i.e.,

$$V_j = \text{span}\, \{\varphi_{jk}(t) := \varphi(2^j\,t - k),\, k \in \mathbb{Z},\, t \in [0, \infty)\}, \quad j \geq 0.$$

Let $V_j^0[0, +\infty)$ denote the restriction of $V_j$ to the semi-infinite interval $[0, +\infty)$. Thus, $V_j^0[0, +\infty)$ is generated by suitable functions $\varphi_{jk}$, i.e., with $\varphi_{jk}(0) = 0$:

$$V_j^0[0, +\infty) = \text{span}\, \{\varphi_{jk}(t),\, k \in \mathbb{Z},\, t \in [0, \infty]\}, \quad j \geq 0.$$

It is possible to prove that $V_j^0[0, +\infty)$ still generate an MRA on $[0, +\infty)$ [22].

### 5. The Fractional Derivative of B-Splines

In this section, we want to analyse the application of the fractional relation (5) to the B-splines in $I = [0, \infty)$, in particular for the left-edge functions. Let $[0, n + 1]$ be the support of the cardinal B-spline $B^n$. Let us consider

$$B_k^n(x) := B^n(x - k).$$

Note that, for the $B_k^n$ whose support is entirely contained in $I$, the index $k$ is greater than or equal to zero, while the left-edge B-splines have index $k = -n, \ldots, -1$. Thus, the following two theorems state the adapted differentiation relation for both types. First, we state a theorem for the exact calculation of some integrals involved in the integration of the B-splines derivative. By (6), we have

$$B_0^n(x) = \frac{1}{n!} \sum_{r=0}^{n+1} (-1)^r \binom{n + 1}{r} (x - r)_+^n$$

and for the derivative,

$$\frac{\mathrm{d}}{\mathrm{d}x} B_0^n(x) = \frac{1}{(n - 1)!} \sum_{r=0}^{n+1} (-1)^r \binom{n + 1}{r} (x - r)_+^{n-1}.$$

If the index $k = -n, \ldots, -1$, we have the left-edge B-splines. To determine the fractional derivative of these functions, for $x > 0$ and $-n \leq k \leq -1$, we use the difference

between the whole integral $_{-k}D_x^\alpha B_k^n(x)$, which is known, and the part before zero, $_0D_x^\alpha B_k^n(x)$:

$$_0D_x^\alpha B_0^n(x) := \frac{1}{\Gamma(1-\alpha)} \int_0^x \frac{B_k'(t)}{(x-t)^\alpha} dt =_{-k} D_x^\alpha B_k^n(x) -_{-k} D_0^\alpha B_k^n(x) \tag{7}$$

where

$$_{-k}D_x^\alpha B_k^n(x) := \frac{1}{\Gamma(1-\alpha)} \int_{-k}^x \frac{B_k'(s+k)}{(x-k-s)^\alpha} ds.$$

By summing and subtracting the index $k$ and using the change of the variable $t - k = s$, we obtain the following equation:

$$_kD_0^\alpha B_k^n(x) = \frac{1}{\Gamma(1-\alpha)} \int_k^0 \frac{B_k'(s+k)}{(x-k-s)^\alpha} ds =$$

$$= \frac{1}{\Gamma(1-\alpha)} \int_0^{-k} \frac{B_0'(s)}{(x-k-s)^\alpha} ds.$$

Therefore, (7) can be rewritten as

$$_0D_x^\alpha B_k^n(x) = \left.\frac{\Delta^{n+1}x_+^{n-\alpha}}{\Gamma(n+1-\alpha)}\right|_{x-k} -$$

$$- \frac{1}{\Gamma(1-\alpha)} \int_0^{-k} \frac{B_0'(s)}{(x-k-s)^\alpha} ds \quad k = -n, \ldots, -1.$$

The calculus of the fractional derivatives can be reduced to evaluating $_kD_0^\alpha B_k^n(x)$.

**Theorem 2.** *For $0 \le \alpha \le 1$, for the B-splines with all the support in $I = [0,T]$, one has*

$$_kD_x^\alpha B_k^n(x) =_0 D_{x-k}^\alpha B_0^n(x-k), \quad k = 0, \ldots, b-a,$$

*while for the left B-splines,*

$$_0D_x^\alpha B_k^n(x) = \left.\frac{\Delta^{n+1}(x)_+^{n-\alpha}}{\Gamma(n+1-\alpha)}\right|_{x-k} - \frac{1}{\Gamma(1-\alpha)} \int_0^{-k} \frac{B_0'(s)}{(x-k-s)^\alpha} ds, \quad k = -n, \ldots, -1$$

*whereby the second term has the following recurrence relation:*

$$\frac{1}{\Gamma(1-\alpha)} \int_0^{-k} \frac{B_0'(s)}{(x-k-s)^\alpha} ds =$$

$$= \frac{1}{\Gamma(n+1-\alpha)} \sum_{r=0}^{-k-1} (-1)^r \binom{n+1}{r} \left[ (x-k-r)^{(n-\alpha)} + x^{1-\alpha} \right.$$

$$\left. \sum_{p=0}^{n-1} \frac{(-1)^{n-p}(-k-r)^{n-1-p}(x-k-r)^p}{(n-1-p)!} \prod_{s=1}^{n-1-p} (\alpha-s) \right].$$

**Proof.** For the B-splines contained in $[0,T]$, we have

$$_kD_x^\alpha B_k^n(x) = \frac{1}{\Gamma(1-\alpha)} \int_k^x \frac{B_k'(t)}{(x-t)^\alpha} dt.$$

By summing and subtracting $k$ and by setting $t - k = s$, we have

$$_kD_x^\alpha B_k^n(x) = \frac{1}{\Gamma(1-\alpha)} \int_0^{x-k} \frac{B_k'(s+k)}{(x-k-s)^\alpha} ds.$$

Since $B_k'(x) = B_0'(x-k)$, we have

$$_k D_x^\alpha B_k^n(x) = \frac{1}{\Gamma(1-\alpha)} \int_0^{x-k} \frac{B_0'(s)}{(x-k-s)^\alpha}\, ds =_0 D_{x-k}^\alpha B_0^n(x-k);$$

This is the first thesis.

As for the second thesis, we remember that $B_0'(x)$ is written as

$$B_0'(x) = \frac{d}{dx} \frac{\Delta^{n+1} x_+^n}{n!} = \frac{1}{(n-1)!} \sum_{r=0}^{n+1} (-1)^r \binom{n+1}{r} (x-r)_+^{n-1}.$$

Putting it in (7) and setting $s - r = t$, we obtain

$$_{-k} D_0^\alpha B_k^n(x) = \frac{1}{\Gamma(1-\alpha)(n-1)!} \sum_{r=0}^{-k-1} (-1)^r \binom{n+1}{r} \int_{-r}^{-k-r} \frac{t_+^{n-1}}{(x-k-r-t)^\alpha}\, dt$$

$$= \frac{1}{\Gamma(1-\alpha)(n-1)!} \sum_{r=0}^{-k-1} (-1)^r \binom{n+1}{r} \int_0^{-k-r} \frac{t_+^{n-1}}{(x-k-r-t)^\alpha}\, dt$$

which is the sum of only the first $-k$ terms.

The integration rules for rational functions are

$$\int_0^{-k} \frac{t^{n-1}}{(x-k-t)^\alpha}\, dt =$$

$$\frac{(n-1)!}{\prod_{s=1}^n (\alpha-s)} (x-k-t)^{(1-\alpha)} \sum_{r=0}^{n-1} \frac{(k-x)^r (t)^{n-1-r}}{(n-1-r)!} \prod_{s=1}^{n-1-r} (\alpha-s) \Bigg|_{t=0}^{-k} =$$

$$\frac{(n-1)!}{\prod_{s=1}^n (\alpha-s)} \Bigg[ (x-k)^{(n-\alpha)} + x^{1-\alpha} \sum_{r=0}^{n-1} \frac{(-1)^{n-r} (-k)^{n-1-r} (x-k)^r}{(n-1-r)!} \cdot$$

$$\cdot \prod_{s=1}^{n-1-r} (\alpha-s) \Bigg].$$

$\square$

## 6. The Fractional Collocation Method

Let $j$ be fixed, and for simplicity, we ignore the dependence of $j$.

In the theory, we are looking for a vector of approximating functions in $V_j[0,\infty)$, i.e.,

$$\tilde{x}_i(t) = \sum_{k\in\mathbb{Z}} c_{ik}\, \varphi_{jk}(t), \quad i = 1,\dots,n$$

which solves the differential problem (1) on a set of *collocation points*.

In practice, since $\varphi_{jk}$ have a fast decay, the series reduces to a finite number of terms:

$$x_i(t) = \sum_{k=K_1}^{K_2} c_{ik}\, \varphi_{jk}(t), \quad i = 1,\dots,n \tag{8}$$

where $K_1$ and $K_2$ are given by

$$K_1 = \min\{k \in \mathbb{Z} \mid \operatorname{supp}(\varphi_{jk}) \cap [0,1] \neq \varnothing\}$$

and

$$K_2 = \max\{k \in \mathbb{Z} \mid \operatorname{supp}(\varphi_{jk}) \cap [0,1] \neq \varnothing\}.$$

Let

$$
C = \begin{pmatrix} c_{1K_1} & \cdots & c_{1K_2} \\ \vdots & \ddots & \vdots \\ c_{nK_1} & \cdots & c_{nK_2} \end{pmatrix}
$$

be the coefficient matrix and

$$
\Phi(t) = \begin{pmatrix} \varphi_{jK_1}(t) \\ \vdots \\ \varphi_{jK_2}(t) \end{pmatrix}.
$$

Therefore, we can rewrite (8) in matrix form:

$$
X = C\Phi(t). \tag{9}
$$

Using the linearity of $D^\gamma$ and (8), we obtain

$$
D^\gamma X(t) = CD(t) \tag{10}
$$

where

$$
D(t) = \begin{pmatrix} D^\gamma \varphi_{jK_1}(t) \\ \vdots \\ D^\gamma \varphi_{jK_2}(t) \end{pmatrix}.
$$

If we choose as collocation points the *dyadic nodes* that are defined by $\{t_p = p/2^s, \ p = 0, \ldots, 2^s\}$, then

$$
\begin{cases} D^\gamma X(t_p) = A(t_p)\, X(t_p), & p = 0, \ldots, 2^s \\ X(0) = X_0. \end{cases} \tag{11}
$$

By (9) and (10), (11) becomes

$$
\begin{cases} CD(t_p) = A(t_p)C\Phi(t_p), & p = 0, \ldots, 2^s \\ C\Phi(0) = X_0. \end{cases}
$$

Using the Kronecker product, we obtain the following linear algebraic system:

$$
\begin{cases} \left(D(t_p)^T \otimes I_n - \Phi(t_p)^T \otimes A(t_p)\right) \mathrm{vec}(C) = 0, & p = 0, \ldots, 2^s \\ \left(\Phi(0)^T \otimes I_n\right) \mathrm{vec}(C) = X_0. \end{cases}
$$

where $\mathrm{vec}(X)$ denotes the vectorisation of the matrix X, formed by stacking the columns of X into a single column vector.

In the next section, we will solve a specific test problem numerically.

## 7. Numerical Results

In the test, we used the cubic B-spline $B^3$ as an approximation function.

We set

$$
\tilde{\varphi}_{jk}(t) := 2^{\frac{j}{2}} B^3(2^j t - k).
$$

Therefore, we define the spaces:

$$
V_j[0,1] = \overline{\mathrm{span}\left\{\tilde{\varphi}_{jk}(t), k \in \mathcal{N}_j, t \in [0,1]\right\}} \quad j \geq j_0
$$

where $\mathcal{N}_j \subset \mathbb{Z}$ is the set of admissible indices $k$ and $j_0$ is the initial multiresolution scale.

It is not difficult to prove that these spaces produce an MRA on $[0,1]$. See [25] for more details.

Now, we consider the following example of a fractional dynamical system in two dimensions:

$$\begin{cases} D^\gamma x(t) = 2x(t) - y(t) \\ D^\gamma y(t) = 4x(t) - 3y(t), & t \in [0,1] \\ x(0) = 1.2 \\ y(0) = 4.2 \end{cases} \tag{12}$$

where $0 < \gamma < 1$ and whose exact solution is [31]

$$\begin{cases} x(t) = \frac{1}{5} E_\gamma(t^\gamma) + E_\gamma(-2t^\gamma) \\ y(t) = \frac{1}{5} E_\gamma(t^\gamma) + 4E_\gamma(-2t^\gamma) \end{cases}.$$

$E_\gamma$ is the Mittag–Leffler function, i.e.,

$$E_\gamma(z) = \sum_{k=0}^\infty \frac{z^k}{\Gamma(\gamma k + 1)}.$$

We note that the matrix associated with the dynamical system has eigenvalues $\lambda_I$ such that $\min |\arg \lambda_i| > \gamma \pi / 2$, so that the stability of the system is guaranteed [32].

Using the collocation method with $s = 8$ and $j = 8$, we numerically solved the dynamic system in the interval $[0,1]$, and we compared it with the exact solution (Figure 2). As expected, the error is of the order of the machine precision.

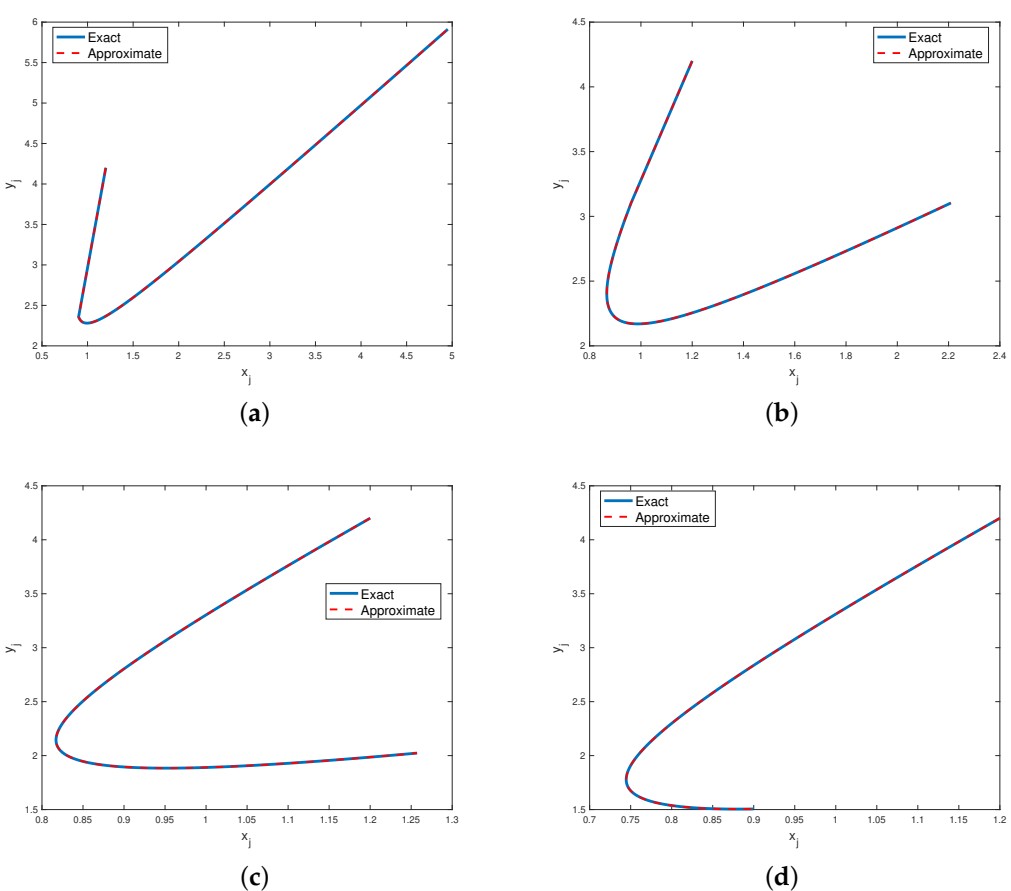

**Figure 2.** The approximate solutions $x_j$, $y_j$ with $j = 8$ (red line) obtained with the cubic B-spline $\varphi_{3,3} \equiv B_3$ and the exact solutions $x(t)$, $y(t)$ (blue dashed line). We consider four different example obtained whit different value for $\gamma$. In (**a**) we use $\gamma = 0.10$, in (**b**) $\gamma = 0.25$, in (**c**) $\gamma = 0.5$ and in (**d**) $\gamma = 0.75$.

## 8. Conclusions and Future Work

We constructed a collocation method that uses fractional refinable functions as approximate functions, to solve a fractional non-stationary dynamical system. We provided an explicit formula that allowed us to evaluate the fractional derivatives of the approximate functions in an accurate and simple way. This formula uses a linear combination of fractional refinable functions of minor order to obtain the prescribed derivative. The method can be efficiently implemented by using standard multiscale techniques to evaluate the fractional B-splines and least mean squares for the final rectangular system. The numerical results showed that this collocation method approximates the solution of a test fractional system with good accuracy. In the future, we can use, as approximation functions, the fractional refinable functions described in [29]. The central point of the work is that this efficient technique can be extended to any fractional differential problem.

**Author Contributions:** Conceptualization, L.P. and S.D.L.; methodology, L.P. and S.D.L.; software, S.D.L.; writing—original draft preparation, L.P. and S.D.L.; writing—review and editing, L.P. and S.D.L.; funding acquisition, L.P. All authors have read and agreed to the published version of the manuscript.

**Funding:** The APC was payed using the fund "00300_22_RS_PEZZA_SEED_PNR_2022-PEZZA-ATENEO SEED PNR 2022".

**Conflicts of Interest:** The authors declare no conflict of interest. The funders had no role in the design of the study; in the collection, analyses, or interpretation of data; in the writing of the manuscript; or in the decision to publish the results.

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
