# Peer review of "Fractional Dynamical Systems Solved by a Collocation Method Based on Refinable Spaces"

_axioms, doi:10.3390/axioms12050451_

Round 1

Reviewer 1 Report

Please see the attached report.

Reviewer 2 Report

1.      The paper is not in the required journal form.

2.      The abstract is not a brief description of the given paper.  

3.      Is this paper already submitted to the other journal? (at the first page is written: Preprint submitted to Elsevier)

4.      The introduction does not overview similar work in the research field. The paper’s state of the art is not well stated.

5.      The abbreviation GP is not defined

6.      The numerical results are too short no relevant comment regarding the example and other computational parameters, such as approximation function, interval, etc.

Round 2

Reviewer 1 Report

Most of the suggested changes has been incorporated in the revise version.
